# Recent Advances in Synthetic Routes to Azacycles

**DOI:** 10.3390/molecules28062737

**Published:** 2023-03-17

**Authors:** Anh Thu Nguyen, Hee-Kwon Kim

**Affiliations:** 1Department of Nuclear Medicine, Jeonbuk National University Medical School and Hospital, Jeonju 54907, Republic of Korea; 2Research Institute of Clinical Medicine of Jeonbuk National University-Biomedical Research Institute of Jeonbuk National University Hospital, Jeonju 54907, Republic of Korea

**Keywords:** azacycles, *N*-heterocycles, heterocyclic synthesis

## Abstract

A heterocycle is an important structural scaffold of many organic compounds found in pharmaceuticals, materials, agrochemicals, and biological processes. Azacycles are one of the most common motifs of a heterocycle and have a variety of applications, including in pharmaceuticals. Therefore, azacycles have received significant attention from scientists and a variety of methods of synthesizing azacycles have been developed because their efficient synthesis plays a vital role in the production of many useful compounds. In this review, we summarize recent approaches to preparing azacycles via different methods as well as describe plausible reaction mechanisms.

## 1. Introduction

Heterocyclic compounds are frequently identified and play an important role in human life due to their special structures. For example, heterocyclic structures are related to many biological processes and form the basic skeleton of many drug molecules and natural products [1,2,3,4,5,6,7,8,9,10,11,12,13,14,15,16,17]. *N*-heterocycles, which contain nitrogen atoms, have attracted much attention from scientists because of their unique properties and diverse utilization. *N*-heterocycles have been employed in many industries, including as dyes, agrochemicals, and materials [18,19,20,21,22,23,24,25,26,27,28,29]. In pharmaceuticals, small-molecule drugs contain nitrogen-containing heterocycles and exhibit diverse bioactivities including anti-Alzheimer’s, antivirus, and anticancer behavior [30,31,32,33,34,35,36,37,38,39,40,41]. Thus, a series of studies on the synthesis and functionalization of many *N*-heterocyclic compounds, such as indoles, imidazoles, pyrrolidines, indolizines, and quinolines, as well as their application, has been carried out.

Azacycle, a nitrogen-containing heterocycle, is an important scaffold in *N*-heterocycles. Statistically, more than half of the small-molecule drugs approved by the United States Food and Drug Administration (FDA) contain azacycle skeletons [1], and numerous drugs on the market share a similar azacycle moiety. For instance, captopril is an important medicine for the treatment of hypertension, pibrentasvir is an antiviral agent for the treatment of hepatitis C, gilteritinib was approved by the FDA for the treatment of relapsed or refractory acute myeloid leukemia with a FLT3 mutation, and futibatinib was recently approved by the FDA for the treatment of metastatic intrahepatic cholangiocarcinoma (Figure 1) [42,43,44,45]. Due to their enormous potential, the synthesis of *N*-substituted heterocycle building blocks is a valuable challenge in organic and medicinal chemistry. Historically, numerous attempts have been made to synthesize azacycles. Several studies have succeeded in synthesizing or functionalizing azacycle compounds. In addition, several methods of synthesis for aromatic azacycles have been reported [46,47,48,49,50,51,52,53,54,55,56,57,58,59,60]. However, these methods still have several drawbacks such as being time-consuming, requiring a high temperature, expensive additives, and/or organic solvents, and/or having low chemoselectivity properties [61,62,63]. 

In recent years, many researchers have developed novel approaches to forming azacycle molecules by designing more effective, convenient, economical, and green processes. The present review summarizes recent advances in the synthesis of the following azacycles: azetidine, pyrrolidine, piperidine, azepane, etc. 

## 2. Reactions

### 2.1. Dialkylation of Primary Amines with Dihalides

One of the old synthetic protocols for the preparation of azacycles is the reaction of amines with dihalides (Br and Cl), reported by Hill and co-workers in 1954 [64]. A microwave reaction was also developed as a useful method for the synthesis of azacyclic compounds. Before 2010, a series of reactions using microwaves between amines and dihalides was reported, and these microwave-assisted syntheses were usually achieved within 20 min at 110–150 °C [65,66,67,68,69].

In 2007, Patel and coworkers performed *N*-alkylation of anilines with halides in the presence of sodium dodecyl sulfate (SDS) and NaHCO_3_ in H_2_O for the preparation of azacycle compounds (Figure 1) [70]. In aqueous-mediated *N*-alkylation of amines, a variety of six- and seven-membered *N*-aryl heterocyclic amines were synthesized from aniline’s derivatives and alkyl dihalides via alkylation and intramolecular cyclization. Several aniline derivatives with both electron-donating and electron-withdrawing groups were successfully employed in this reaction, providing desired products (**3a**–**3f**) with good yields.

Another microreactor system was employed for the synthesis of azacycles by Gao and co-workers (Figure 2) [71]. In order to overcome the uncontrollable local temperature inside a conventional reaction batch, they used a microreactor system with separate pumps to inject each precursor into a micromixer in precise order and amounts. Reactions of aniline’s derivatives with halides in the presence of K_2_CO_3_ in a water–ethanol solvent mixture were carried out at 120 °C and 75 psi. Controlling the residence time through adjusting the flow rate to increase retention time leads to the formation of products within 5 min. Various functional groups on aniline were tolerated with the reaction protocol using a microreactor. A reaction using aniline’s derivatives bearing electron-donating groups and electron-withdrawing groups with a longer retention time was smoothly conducted to produce azacycles (**6a**–**6d**) with a five-membered ring at high yields. Additionally, azacycles (**6e**–**6g**) with six- and seven-membered rings were successfully formed with good yield (over 60%) using this microreactor system. However, a four-membered ring product (**6h**) was obtained with 30% under the same reaction conditions due to significant ring strain. In addition, the synthesis of ester-substituted azacycles was investigated. It was found that these ester groups are often hydrolyzed during the cyclization reaction of amines in basic conditions. However, a retention time of 5 min resulted in high selectivity in the alkylation of amines. It was explained that the ability to generate heat and transfer precursor rapidly helped to form the product more quickly under basic conditions. 

### 2.2. N-Heterocylization of Primary Amines with Diols

In 2013, Shi and co-workers developed metal-catalyzed double *N*-alkylation of amines with diols for synthesizing azacycles [72]. The reactions between amines and alcohols were conducted in the presence of NiCuFeO_x_ catalyst in xylene at reflux for 24 h (Figure 3). A broad range of amine sources, including aromatic and aliphatic primary amines, secondary amines, and ammonia, were tolerated with this reaction, affording azacycles (**9a**–**9e**) with good yields. In this study, different types of diols were successfully used in the process to form five-, six-, or seven-membered *N*-heterocycles (**9f**–**9h**) with good yield (73–93%).

Ni-catalyzed synthesis of *N*-heterocycles, including azacyles from amines and diols, were reported by Tang and co-workers in 2019 [73]. The processes employed Ni(OTf)_2_ and 1,2-bis(dicyclohexylphosphino)-ethane (dcype) as catalysts to transfer hydrogen and were conducted in 1,1,1,3,3,3-hexafluoroisopropanol (HFIP) as a solvent at 120 °C (Figure 4). The reaction worked effectively on several anilines bearing electron-donating groups, providing *N*-aryl piperidines (**12a**–**12c**) with high yields. Additionally, five- and seven-membered azacycles (**12d**–**12f**) were also synthesized with moderate to good yields using the same process. 

In 2020, Donohoe and co-workers developed an iridium-catalyzed annulation reaction between primary amines and diols [74]. The reaction of amines with diols was carried out in the presence of [Cp*IrCl_2_]_2_ in water at 80 °C (Figure 5). In this study, they attempted to design an enantioselective reaction to reduce racemization. This reaction protocol was applied for the synthesis of monosubstituted *N*-benzyl piperidines at the C3 and C4 positions. Various diols were successfully employed in this annulation reaction. In particular, diols bearing aliphatic, aromatic, and bulky groups, as well as diols with electron-donating and electron-withdrawing groups at the C4 and C3 positions, were well tolerated with this reaction, affording products (**15a**–**15d)** in high yield with excellent diastereoselectivity. Diols with a heteroatom in the skeleton were also tested and readily yielded morpholine. Several multi-substituted heterocycles with substituent positions and stereo configurations were effectively employed for this process to produce the *N*-heterocycle (**15e**) with preserved absolute stereochemistry. In addition, this reaction also occurred in the synthesis of bicyclic azacycle **15f** with good yield (75%). A variety of substituted amines with electron-rich and electron-poor groups as well as steric hindrance groups was also well tolerated in the process, affording products (**15g**, **15h**) with good yields (62–86%). The annulation reaction of various substituents on amine with a certain diol isomer smoothly afforded the desired products (**15i**–**15n**) with moderate to good yields while preserving the absolute configuration of the chiral carbon. 

### 2.3. N-Heterocylization of Primary Amines with Dicarbonyl Compounds

#### 2.3.1. *N*-Heterocylization of Primary Amines with Dialdehydrides

Dialdehydrides have been used for the reaction of amines to produce azacyclic compounds. A series of reactions of dialdehydrides with amines to produce azacycles was reported before 2000 [31,32,33,34]. Most of these processes were achieved via reductive amination of aldehydes. 

In 2000, Baba and co-workers reported the reductive amination of aldehydes and amines using a tin hydride system for the synthesis of azacycles [75]. The reductive amination reactions were carried out in the presence of Bu_2_SnClH-HMPA in THF at −78 °C or 0 °C (Figure 6). In the process, dialdehydes were successfully treated with primary amines to produce *N*-substituted cycle amines including azacycles (**18a**, **18b**, **21**) with good yields (63–74%). The reaction scope was also expanded to the reaction of amino esters and amino alcohols, which resulted in products with good yields. However, aliphatic amines such as isopropyl, benzyl, and other alkyl amines were not well tolerated with this method due to their strong basicity. 

A plausible mechanism for this process, proposed by Baba and co-workers, is depicted in Figure 7. Carbonyl compound **22** was reacted with amine to form imine **23** and then tin chloride reductant reacted with **23** to form an iminium salt complex, **25**. It was proposed that the long Sn–Cl bond provided an easy way to form iminium salts. The charged iminium salt made it more easily reduced by hydride than other reducible groups like carbonyl or multiple bonds, thus leading to the high selectivity of the process. Finally, **25** was converted to **26**. The reduced complex **26** was then reacted with hydrogen ions to generate amine **27**, and the tin chloride complex was returned.

#### 2.3.2. *N*-Heterocylization of Primary Amines with Dicarboxylic Acids

Carboxylic acids were also employed for the synthesis of azacycles. In 2020, Darcel and co-workers carried out the synthesis of *N*-substituted cyclic amine from diacids and amines via a hydrosilylation reaction [76]. The hydrosilylation reactions between diacid and amine were achieved via a reaction in the presence of Fe(CO)_4_(IMes) as a catalyst, Fe(OTf)_2_ as an additive, and phenylsilane in DMC at 110 °C (Figure 8). Various functional groups, including electron-donating and electron-withdrawing groups and heteroaromatic groups on aliphatic amine, were well tolerated in the reaction, affording azacycles (**34a**–**34d**) with good yields (68–96%). However, when bulky group-substituted amines were employed, the reaction yield was reduced due to the effect of steric hindrance on the reaction (**34e**). Reactions using halogen- and electron-donating group-substituted anilines smoothly afforded the azacycle products (**33b**–**33d**, **34f**, **34g**, **35c**) with moderate to good yields (50–95%). However, reactions using electron-withdrawing groups such as nitro or cyano did not yield the desired products. Interestingly, reactions of aniline-substituted alkyl amine substrates with diacids showed high selectivity for *N*-alkylation of aliphatic nitrogen, producing the desired products (**34h**, **33e**, **35d**) with 85–91% yields, while the aniline moiety remained unaffected. Additionally, useful drugs (Fenpiprane and Prozapine) were readily prepared with good yields via this reaction method. 

A probable mechanism of the reaction is presented in Figure 9. Dehydrogenative silylation of diacid **28** formed silylated diester **36** and generated H_2_. Removal of R_3_Si-O-SiR_3_ provided a cyclic anhydride, **37**, which may undergo reduction to form diol **39** but primarily reacts with amine to form an intermediate imide, **38**. This imide was then reduced to amide **40**, which was converted to cyclic amine **33** via hydrosilylation catalyzed by the iron complex. 

In 2022, Kim and co-workers reported SnCl_2_-catalyzed reductive amination between dicarboxylic acids and aryl amines for the synthesis of azacycles [77]. Reactions between aniline’s derivatives and diacids were carried out in the presence of SnCl_2_ and PhSiH_3_ in toluene at 110 °C (Figure 10). The SnCl_2_-catalyzed reactions successfully produced various *N*-aryl cyclic amines bearing a five-membered ring moiety (**43a**–**43d**). A wide range of substituents on aniline, including electron-donating groups such as methoxy, ethyl, and *tert*-butyl groups, and electron-withdrawing groups such as halogens and nitrile groups, was tolerated with the reaction, leading to the generation of the corresponding products (**43e**–**43h**) with good yields (66–87%). Additionally, reactions using adipic acid readily afforded seven-membered azacycles (**43i**, **43j**) with high yields (75–85%). Moreover, using this method, more complex azacycles such as *N*-aryl isoindolines and *N*-aryl tetrahydroisoquinolines (**43k**, **43l**) were successfully synthesized with high yields. 

The proposed pathway for this reaction is shown in Figure 11. Initially, succinic acid **42** was dehydrogenatively silylated by phenylsilane, producing diester **44** and H_2_. Release of **45** resulted in the formation of a cyclic anhydride, **46**, which reacted with aniline to form 1-phenylpyrrolidine-2,5-dione **47**. Two reductions of **47** in the presence of PhSiH_3_ and SnCl_2_ generated target product **43**. 

#### 2.3.3. *N*-Heterocyclization of Primary Amines with Diesters 

In 2017, Harvie and co-workers developed the synthesis of azacycles from diesters via hydrogenation [78]. The reactions of 1,6-hexanedioate with aniline were carried out in the presence of [Ru(acac)_3_]_,_ triphos, and methanesulfonic acid (MSA) as catalysts and hydrogen gas in dioxane at 220 °C for 42 h (Table 1). Using this method, alkyl, aryl, and bulky alkyl esters were readily converted to *N*-heterocycle products (Table 1, entries **1**–**3**), while reactions using diacids did not efficiently yield the target products (Table 1, entry **4**). Moreover, various five-, six-, seven-, and eight-membered azacycle products (Table 1, entries **5**–**7**) were prepared from this reaction process with good yields (66–92%). The reaction using branched diester also afforded the corresponding products (Table 1, entry **8**) with good yield. However, both pure enantiomers were racemized after reaction (Table 1, entries **9**, **10**). 

The proposed pathway of this reaction is shown in Figure 12. Hydrogenation of one ester group of substrate **49** formed ester aldehyde **52**. This ester aldehyde reacted with aniline to provide an imine, which was reduced by H_2_ to afford compound **53**. The remaining ester group of **53** was also hydrogenated to give aldehyde **54**, which underwent cyclization to form *N*-heterocycle **51**. 

### 2.4. N-Heterocyclization of Primary Amines with Cyclic Ethers

#### 2.4.1. Metal-Based *N*-Heterocyclization of Primary Amines with Cyclic Ethers

Reactions of amine with cyclic ethers were carried out for the synthesis of azacyclic compounds. A series of reactions using metal-based reagents including alumina, Al_2_O_3_, and TiO_2_ was reported before 2000 [79,80,81]. In 2014, Lee and co-workers reported AlMe_3_ mediated synthesis of *N*-aryl *N*-heterocycles from cyclic ethers and aniline derivatives in toluene at 110 °C (Figure 13) [82]. Reactions of tetrahydrofuran (THF) with a wide range of aromatic amines bearing electron-donating groups successfully afforded azacycles (**57a**–**57c**) with 70–72% yields. Reactions using aromatic amines with electron-withdrawing groups like chloride, fluoride, and bromide yielded the corresponding products (**57d**, **57e**) with increased reaction yields. 2-Methyltetrahydrofuran and 4-fluoroaniline were smoothly employed in this reaction to prepare azacycle **57f**, with 90% yield. Compound **57g** containing napthyl was also synthesized, with moderate yield. In addition, tetrahydropyran was tolerated with the reaction to provide *N*-aryl piperidine (**57h**), with good yield. Several fused heterocyclic systems including tetrahydroisoquinilines and isoindolines were also prepared by conducting this reaction in xylene at 150 °C. Aniline and its derivatives bearing electron-donating and electron-withdrawing groups were readily used for the process to give fused heterocyclic compounds (**57i**–**57n**). 

A probable mechanism was suggested by Lee and co-workers (Figure 14). Control experiments showed that the formation of compound **57** via the transformation of compound **61** in the presence of AlMe_3_ was achieved to support the mechanism. Reaction of aniline and AlMe_3_ generated dimethyl aluminum-amide **58** and methane. Then, THF was added to **58** to form complex **59**. Later, attack of nucleophilic amide at the α-carbon of tetrahydrofuran **59** provided cycle **60**. The amide of **60** attacked the other carbon at the α position to oxygen, resulting in the formation of azacycle **57**. 

In 2015, Poliakoff and co-workers developed a self-optimizing continuous-flow reaction involving aniline, dimethyl carbonate (DMC), and THF in the presence of supercritical CO_2_ and γ-Al_2_O_3_ at high pressure (10–20 MPa) (Figure 15) [83]. The reaction between aniline and THF in the presence of DMC generated *N*,*N*-dimethylaniline **63** as the major product, as well as several *N*-substituted byproducts such as *N*-methylaniline **64**, methyl phenylcarbamate **65**, and *N*,*N*-4-trimethylaniline **66**. Remarkably, when the reaction was performed in the absence of DMC, *N*-phenylpyrrolidine **68** was found to be the predominant product with an over 99% yield. 2-Methyltetrahydrofuran was also tolerated with this cyclization reaction, and compound **70** was smoothly produced.

A possible pathway for the synthesis of *N*-aryl cyclic amine is presented in Figure 16. The nucleophilic nitrogen of aniline attacked THF, generating an amino alcohol intermediate **72**. In the absence of DMC, intermediate **72** underwent an intramolecular nucleophile substitution, leading to the formation of the desired compound **68**. However, in the presence of DMC, the labile amino alcohol **72** would be alkylated by DMC to form compound **73**. The alkylation of **73** by another DMC and THF produced byproducts **74** and **75**, respectively. 

In 2017, Wang and co-workers reported the transformation of aniline and cyclic ethers to *N*-aryl azacycles in the presence of TiCl_4_ in toluene at 110 °C for 24 h (Figure 17) [84]. Reactions of aniline and its halogen derivatives produced the corresponding products (**78a**, **78b**) with 68–76% yields. Electron-withdrawing groups such as nitro and electron-donating groups such as the methyl group on aromatic amines were tolerated with this reaction, which gave target products (**78c**, **78d**) with 59% and 60% yields, respectively. In addition, 2-Methyltetrahydrofuran was also employed in the process to generate the product (**78e**) with 67% yield. Using this method, tetrahydropyran was effectively converted to *N*-aryl six-membered azacycle (**78f**) in xylene at 140 °C. The reaction scope was further expanded to successfully synthesize fused *N*-heterocycles (**78g**, **78h**). 

A plausible mechanism was suggested via calculation of the Gibbs free energies (Figure 18). The kinetic study of the reaction between 4-fluoroaniline and THF suggested a pseudo-first-order reaction with a rate constant of 5 × 10^−5^ s^−1^ and an activation energy of 30 kcal mol^−1^. This activation energy was consistent with the required energy of the proposed mechanism (26.9 kcal mol^−1^). The reaction of aniline, TiCl_4_, and THF formed complex **79**. Calculation of the Gibbs free energies showed that the rate-determining step was the ring opening of the activated cyclic ether. Nucleophilic attack of the nitrogen of **79** on the α-carbon of the activated THF ring formed transition state **80**. The cyclic ether ring of **80** was opened to yield **81**, followed by HCl elimination and the formation of a seven-membered ring to give **82**. The α-carbon of oxygen in **82** was subsequently attacked by nucleophilic nitrogen to form **83** with a new C–N bond. Ring closing generated the azacycle product **78** and a titanium complex. 

Reaction of *N*-alkyl-protected arylamines with THF in the presence of TiCl_4_ and 1,8-diazabicyclo[5.4.0]undec-7-ene (DBU) for the synthesis of azacycles was reported by Kim and co-workers in 2020 (Figure 19) [85]. Reaction between *N*-alkyl-protected arylamine with THF in the presence of only TiCl_4_ gave the desired product with a lower yield (24%). Thus, several bases were screened to increase the reaction yield and DBU was proven to be an effective base to provide azacycles with high yields. A wide range of *N*-ethyl anilines bearing electron-donating groups and electron-withdrawing groups was effectively transformed into the corresponding *N*-aryl azacycles (**86a**–**86g**) in high yields (70–94%). In addition, steric hindrance did not have any significant negative effects on the reaction yield and compound **86h** was synthesized at a yield of 71% under the same reaction conditions. Reactions of 2-methyltetrahydrofurans with electron-rich and electron-poor arylethyl amines readily produced desired *N*-aryl azacycles (**86i** and **86j**). Tetrahydropyran was also well tolerated in the reaction, affording a six-membered azacycle **86k** with high yield. Using the process, fused ring cyclic ethers such as 1,3-dihydroisobenzofuran and isochromane were successfully transformed to azacycle products (**86l**–**86n**) with no significant effect of the substituents on the benzene ring.

Arylamines protected by various alkyl groups such as methyl, isopropyl, and *tert*-butyl were tolerable in this process and the formation of target *N*-aryl azacycle products (**89a**–**89e**) was achieved with 82–93% yields (Table 2). However, reactions using *N*-alkyl-protected aliphatic amine were not successfully carried out. 

The proposed mechanism of this reaction is shown in Figure 20. Control experiments showed that reaction with TiCl_4_ alone afforded the desired azacycle with low yield, while the reaction with DBU alone generated no desired product, indicating the essential role of DBU in activating the arylamine to increase reaction yield. *N*-Alkyl arylamine was bound to TiCl_4_ to give the titanium complex **90**, followed by the binding of THF to **90** to form a complex, **91**. During the process, HCl was consumed by DBU. Intramolecular attack of the nucleophilic nitrogen of **91** at an α-carbon of cyclic ether led to the formation of a four-membered ring intermediate **92**. The fused ring system was spontaneously opened, forming a seven-membered ring complex, **94**. Eventually, nucleophilic attack of nitrogen of **94** at the carbon-bearing oxygen generated the desired azacycle, **86**, and CH_3_CH_2_Cl and TiOCl_2_ were discharged. 

In 2022, Kim and co-workers used arylhydrazines to synthesize azacycles. The reactions were performed in xylene with TiCl_4_ and 1,5,7-triazabicyclo[4.4.0]dec-5-ene (TBD) at 120 °C, where a variety of arylhydrazines derivatives were converted into *N*-aryl pyrrolidines (Figure 21) [86]. It is noteworthy that *N*-arylhydrazine was nearly inactive when using solely TiCl_4_. Thus, the employment of TBD plays an important role in enhancing the efficiency for the synthesis of azacycles from *N*-arylhydrazines. Many different functional groups such as electron-donating alkyl groups, electron-withdrawing halogen, and steric-hindered groups on aryl hydrazines were well tolerated in the process, affording the corresponding products (**98b**–**98e**) with high yields (77–92%). In this reaction, 2-methyltetrahydrofuran, a sterically hindered cyclic ether, and tetrahydropyran were effectively transformed into the desired products (**98f**–**98h**) with 85–94% yields. Additionally, various *N*-aryl isoindolines were successfully prepared in high yields using this method. In particular, electron-rich and electron-poor *N*-aryl hydrazines were tolerable in the reaction, affording the desired products (**98i**–**98k**) with 84–90% yields. *N*-aryl tetrahydroquinolines bearing electron-donating and electron-withdrawing substituents (**98l**–**98n**) were also synthesized in high yields. 

A probable mechanism for this reaction was suggested by Kim and co-workers (Figure 22). Control reactions of phenylhydrazine in the presence of TiCl_4_ and TBD at 120 °C and at room temperature showed that the formation of aniline was only detected at 120 °C. *N*-Aryl hydrazine **96** reacted with TiCl_4_ to discard NH_3_ and provide aniline **99**. At the same time, THF was bound to TiCl_4_ to form complex **100**. Next, the attack of the nucleophilic nitrogen of aniline **99** at the carbon-bearing oxygen of THF generated complex **101**, while HCl was consumed by TBD. An intramolecular nucleophilic attack in **101** led to the formation of product **98**, while another HCl was removed by TBD and TiOCl_2_ was discarded. 

#### 2.4.2. Non-Metal-Based *N*-Heterocyclization of Primary Amines with Cyclic Ethers

In 2016, Sun and co-workers carried out the synthesis of *N*-aryl azacycles via reaction between aromatic amines and cyclic ethers in the presence of B(C_6_F_5_)_3_ and *p*TSA·H_2_O under an argon atmosphere (Figure 23) [87]. A variety of substituted anilines were successfully employed for the synthesis of azacycle compounds. Several electron-withdrawing groups such as nitro and chloro groups were tolerated in the reaction with THF, affording the corresponding azacycles (**104a**–**104c**) with 76–88% yields. Aromatic amines bearing electro-donating groups, however, were less reactive in the process than aromatic amines bearing electron-withdrawing groups (50% for methoxy (**104d**) and 77% for methyl (**104e**) at the *para* position, respectively). Aromatic amines with steric hindrance were also tested and provided products (**104f**) with drastically reduced reaction yields. In addition, the reaction using 1-naphthylamine was successfully conducted, giving product **104h**, and secondary amine *N*-methylaniline was also converted to the desired product **104g** in moderate yield. Various cyclic ethers were examined for this process. Reaction of 2-methyltetrahydrofuran with aniline and chloroaniline produced the corresponding azacycles (**104i**, **104j**) at lower yields than those of THF. Remarkably, using this reaction, 1,3-dihydroisobenzofuran was smoothly converted to fused cyclic amine **104k** with a 77% yield. However, the use of tetrahydropyran did not give the target product under the same reaction conditions. 

A possible mechanism for the reaction was proposed by Sun and co-workers (Figure 24). Aniline was bound with B(C_6_F_5_)_3_ to form species **105**, which was confirmed by isolating and elucidating its structure with crystal X-ray and NMR. This species, **105**, then reacted with THF to give an isolable adduct, **106**. Elimination of B(C_6_F_5_)_3_ from **106** provided intermediate **107**. Intramolecular annulation occurred to form *N*-aryl cyclic amine **104**, with the aid of pTSA·H_2_O, releasing water in the process.

In 2017, Wang and co-workers synthesized azacycles in the presence of BF_3_·Et_2_O as a Lewis acid mediator in xylene (Figure 25) [88]. Several arylamines bearing electron-withdrawing groups were tolerated with this reaction, providing the corresponding azacycle products (**110a**–**110d**) with moderate yields (47–59%). However, steric hindrance influenced the reaction efficiency and azacycle **110e** was prepared at a reduced yield. Reaction using arylamines bearing an electron-donating group such as 4-methyl aniline did not yield desired products (**110f**). 

A possible mechanism of this reaction was proposed based on the calculation of the Gibbs free energies (Figure 26). The energy profile of this reaction was similar to TiCl_4_-mediated reaction and its activation energy (25.7 kcal mol^−1^) was comparable to that of TiCl_4_-mediated reaction (26.9 kcal mol^−1^) [84]; therefore, a similar mechanism was proposed. However, unlike TiCl_4_-mediated reaction, formation of **114** was the rate-determining step. The reaction between aniline, THF, and BF_3_·Et_2_O formed a Lewis acid–base intermediate complex **111**. Nucleophilic attack of the nitrogen of **111** on the α-carbon of THF of the complex gave complex **112**, which was converted to the seven-membered ring intermediate **113**. Nitrogen then attacked the α-carbon of activated oxygen to give a four-membered ring intermediate, **114**. C–N bond forming and C–O cleavage produced the product complex **115**, which, after elimination of B(O)F_2_, gave the desired product. 

A hydrogen iodide-catalyzed process for the synthesis of *N*-aryl azacycles from aniline’s derivatives and cyclic ethers was reported by Wang and co-workers in 2017 [89]. The reactions were conducted in the presence of hydrogen iodide under a nitrogen atmosphere at 150 °C (Figure 27). A variety of aromatic amines was employed as substrates to react with THF. Electron-donating substituted anilines such as methyl, methoxy, and hydroxy groups were tolerated with this method to afford the corresponding products (**118b**–**118f**). Steric hindrance of substituents at the *ortho* position reduced the reaction efficiency and target products were prepared with decreased reaction yields (66% for 1-(o-tolyl)pyrrolidine **118c** and 46% for 1-(2,6-dimethylphenyl)pyrrolidine **118d**). Similarly, using the process, anilines bearing electron-withdrawing groups, including a fluoro group and amide moiety, were smoothly converted to products (**118g** and **118h**) with 98% and 64% yields, respectively. 

A plausible mechanism for the reaction is illustrated in Figure 28. This mechanism was supported by three facts, including the total inhibition of the reaction by radical inhibitor TEMPO (2,2,6,6-tetramethylpiperidine-1-oxyl), the detection of intermediates **121** and **125**, and a decrease in **125** over the reaction time. Initially, HI was cleaved to generate reactive hydrogen radicals and iodine radicals. Continuously, the iodine radicals reacted with aniline to give resonance-stabilized aminyl radical **119**. Cyclic ether was opened and iodinated by reaction with HI to generate iodine intermediates **120** and **121**. Diiodine **121** then reacted with hydrogen radicals to form radical **122**, which then reacted with radical **119** to form intermediate **123**. On the other hand, intermediate **123** was produced from another reaction chain. Radical **124** was generated by the reaction of **120** with hydrogen radicals and then radical **124** reacted with radical **119** to give intermediate **125**, which reacted with HI to transform to **123**. Finally, cyclization of **123** produced the desired *N*-aryl pyrrolidine **118** and released HI. 

In 2019, Kim and co-workers discovered a non-metal synthetic method for azacycles through a phosphoramidate intermediate [90]. Reactions between arylamines and cyclic ethers were carried out in the presence of POCl_3_ and DBU in xylene at 110 °C (Figure 29). A wide range of electron-donating substituents on arylamines were successfully tolerated with this reaction, producing the corresponding *N*-aryl pyrrolidines (**128a**, **128b**) in high yields. Although 2,6-diisopropylaniline had steric hindrance, reaction of 2,6-diisopropylaniline generated desired azacycle product **128c** with 63% yield. Additionally, reactions of arylamine substrates bearing electron-withdrawing groups such as a nitro group and halogens with THF were smoothly conducted, affording desired products (**128d**). This method tolerates various cyclic ethers such as tetrahydropyran, oxepane, and 1,4-dioxane, and they were efficiently converted to the corresponding *N*-aryl azacycles (**128f**–**128h**) with good yields, suggesting expansion of the reaction scope and applications. Remarkably, steric hindrance of 2-methyltetrahydrofuran had no effect on the reaction efficiency, and target azacycle **128e** was prepared in high yield. In addition, fused ring cyclic ethers also readily reacted with arylamines bearing electron-donating and electron-withdrawing groups, producing *N*-aryl isoindolines (**128i**–**128k**) and *N*-aryl tetrahydroquinolines (**128l**, **128m**). 

A plausible mechanism proposed by Kim and co-workers is depicted in Figure 30. This mechanism was confirmed by several facts obtained from control experiments. Phosphoramidic dichloride **129** was only produced when employing both POCl_3_ and DBU, rather than either alone. In addition, prepared phosphoramidic dichloride was successfully transformed into the desired product in the reaction with THF, which confirmed the formation of **129** during the reaction. The reaction started with the formation of phosphoramidic dichloride **129** in the presence of POCl_3_ and DBU. Nucleophilic attack of nitrogen of **129** allowed the ring opening of THF to generate intermediate **130**. This intermediate underwent an intramolecular nucleophilic substitution to form target azacycle **128**, releasing PO_2_Cl_2_. 

In 2019, Kim and co-workers further examined the POCl_3_-mediated synthesis of *N*-aryl azacycle from *N*-aryl aniline and THF. They proposed a solvent-free synthetic methodology to achieve the reactions (Figure 31) [91]. In solvent-free synthesis using POCl_3_ and DBU, a variety of aniline derivatives and cyclic ether substrates bearing electron-donating and electron-withdrawing groups were smoothly transformed to their corresponding *N*-aryl heterocycles (**134a**–**134h**) with high yields.

### 2.5. C–N Coupling Reaction

Coupling reactions have been used for the synthesis of azacyclic compounds. Common method is a cross-coupling reaction of aryl halides with amines [92,93,94] and various C–N coupling reaction methods have been developed for the production of azacycles.

#### 2.5.1. Coupling Reaction from Cyclic Amines and Hypervalent Iodine Compounds

In 2016, Stuart and Sandtorv used hypervalent iodonium salt for metal-free synthesis of azacycles [95]. They carried out reactions of aryl(TMP)iodonium salts (TMP = 2,4,6-trimethoxyphenyl) with cyclic amines in the presence of KF and water as additives in 1,2-dichloroethane (DCE) at 70 °C to give the corresponding azacycles (Figure 32). Various aryl(TMP)iodonium trifluoroacetates containing electron-withdrawing groups such as trifluoromethyl, nitro, and ester groups were successfully coupled with amines to give morpholines (**137a**–**137c**) with high yields. Reactions using electron-poor aryl bearing two different groups also afforded **137d** with high yield. Moreover, a variety of *N*-heterocycles including six-membered heterocycles (thiomorpholine, piperidine, and piperazine), a five-membered ring (pyrrolidine), and a seven-membered ring (azepane) were used to react with aryl(TMP)iodonium salts, producing the corresponding products (**137e**–**137i**) in high yields (60–93%). 

A proposed mechanism of this reaction is shown in Figure 33. Formations of intermediates diaryliodonium fluoride and aryl fluoride intermediates were not detected by ^19^F NMR. Reaction of diaryliodonium trifluoroacetate salt **135** with cyclic amine **136** generated intermediate **138** via a ligand exchange between TFA and cyclic amine. Subsequently, nitrogen of **138** was coupled to the aryl group to give *N*-aryl azacycle **137**, while I-TMP was eliminated and hydrogen was consumed by base **A**. 

Another coupling reaction using diaryliodonium salt to synthesize azacycles was developed by Olofsson and coworkers in 2018 [96]. Coupling reactions between diaryliodonium salts containing trifluoromethylsulfonyl (OTf) and aliphatic amines were achieved in toluene at 110 °C (Figure 34). *p*-Nitrophenylation of piperidines, pyrrolidine, and tetrahydroquinoline was successfully performed to afford the corresponding azacycle products (**141a**–**141c**) with high yields. In addition, the phenyl group was smoothly coupled to cyclic amines to produce several *N*-phenyl azacycles including *N*-phenyl piperidine **141d**, *N*-phenyl morpholine **141e**, *N*-phenyl thiomorpholine **141f** and 2-methyl-1-phenylindoline **141g**. Additionally, electron-donating groups including *tert*-butyl and methoxy groups were also tolerated in the reaction, producing *N*-aryl cyclic amines (**141h** and **141i**) with moderate yields. 

A mechanism of this process was proposed, as shown in Figure 35. Control experiments showed that this reaction was not affected by adding radical scavenger 1,1-diphenylethylene (DPE), and aryne trap furan, indicating a ligand coupling mechanism. A reversible ligand exchange between OTf of **140** and cyclic amine **139** led to the formation of intermediate **142**. In the presence of a base or excess amine, deprotonation of intermediate **142** gave intermediate **143**. Continuously, the amine of **143** was coupled with an aryl group to generate *N*-aryl azacycle product **141** and released ArI. 

#### 2.5.2. Coupling from Cyclic Amines and Triphenylsulfonium Triflates

In 2018, Zhang and co-workers developed a C–N coupling reaction using triarylsulfonium triflates as a *N*-phenylation agent [97]. The reaction was conducted in the presence of *t*-BuOK or KOH bases under a nitrogen atmosphere at 80 °C (Figure 36). *N*-Phenylation using a variety of primary and secondary amines successfully produced the corresponding azacycle products with good yields. In addition, various *N*-heterocycles were smoothly converted to *N*-aryl heterocycles. Pyrrolidine, piperidine, morpholine, and thiomorpholine were well tolerated with this method, affording azacycle products (**146a**–**146d**) with good to excellent yields. Several fused ring heterocyclic scaffolds such as tetrahydroisoquinoline, phenolthiazine, carbazole, and indoles found in many drugs were also employed using this method to yield azacycles (**146e**–**146i**) with moderate to excellent yields. 

#### 2.5.3. Cross-Coupling Reaction of Secondary Amines and Aryl Compounds

In 2019, Leonori and co-workers reported direct *N*-aryl amination of secondary amines via a visible light-catalyzed N–H/C–H cross-coupling reaction [98]. In this photo reaction process, amines were treated with aromatic compounds in the presence of NCS and Ru(bpy)_3_Cl_2_ in CH_3_CN or hexafluoroisopropanol (HFIP). The addition of HClO_4_ and a basic workup then yielded the desired products (Figure 37). The coupling reactions between piperidine and arene compounds were investigated. Electron-rich benzenes bearing alkyl and alkoxy groups were well tolerated with this method to create products (**149a**–**b**) with good yields. Various functional groups such as protected amine, halide, and trimethyl silane at the *para* position of aromatic rings were successfully employed in this process. Noticeably, the reaction had selectivity for C–H of the ring with higher electron density in compounds bearing two separated aromatic rings or fused rings (**149f** and **149g**). Using this method, a variety of *N*-aryl piperidines such as esters, amines, alcohols, halogens, azide, carbonyl, and sulfonamide at position C3 or C4 (**149h**–**149n**) were successfully synthesized with high yields. Fused *N*-heterocycles and small to medium-sized cyclic amines such as four-, five-, and seven-membered *N*-heterocycles were also converted to the corresponding products (**149o**–**149r**) with good yields under the same reaction conditions.

A proposed mechanism of this method is illustrated in Figure 38. Chlorination of **147** by NCS gave **150**, which then received a proton to generate compound **151**. At the same time, the photocatalyst (PC) Ru(bpy)_3_Cl_2_ was transformed to the excited state (PC*) under irradiation of blue LED. Next, a single electron transfer (SET) process occurred between PC* and compound **151** to afford aminium radical **152** and PC cation radical. Cyclic voltammetry was used to study the redox properties of *N*-chloropiperidine **150**. Upon the addition of HClO_4_, the reduction potential shifted toward positive values, which confirmed the SET reduction of *N*-chloropiperidine **150** upon protonation. The aminium radical **152** reacted with arene to generate intermediate **153**, which further interacted with PC cation radical in another SET process to provide cation **154** and returned the ground state PC. UV-vis absorption studies showed that, when keeping the mixture of **150**, PC, and HClO_4_ in the dark, the mixture absorbed radiation in the blue region, while irradiating the mixture with blue light resulted in a rapid color change from orange to green. Thus, the effect of blue radiation was demonstrated. Finally, cation **154** released one proton to form the intermediate **155**, which would undergo the basic workup to form the final arylated product. 

#### 2.5.4. Cross-Coupling Reaction from Aryl Halides and Amides

In 2021, Tu and co-workers performed a Ni catalyzed cross-coupling reaction between aryl chlorides and amides to give azacycles (Figure 39) [99]. The reaction was conducted in the presence of Ni(COD)_2_ as a catalyst and APr·HCl as the NHC precursor, *t*BuOK as the base, and water in toluene at 35 °C. Phenyl chloride and 4-trifluoromethyl chloride were employed as substrates for the reaction to afford *N*-aryl azacycle products. Cross-coupling reactions of cyclic formamide having different size rings successfully produced *N*-cyclic amines such as pyrrolidine (**158a**), piperidines (**158b**, **158c**), and azepane (**158d**). In addition, heterocyclic formamides readily underwent cross-coupling with aryl chloride to afford the corresponding products including morpholines (**158e**, **158f**) and piperazines (**158g**, **158j**).

The proposed mechanism of this reaction is shown in Figure 40. It was proposed based on several control experiments. In particular, the use of the radical inhibitor TEMPO did not affect the reaction yield, indicating a non-radical reaction mechanism. Furthermore, the detection of R^2^–H by-products confirmed a decarbonylation pathway. Ni(0) reacted with NHC to generate complex **159**. Reaction of **159** with aryl chloride generated an intermediate **160**. In addition, decarbonylation of amide substrate **157** by *t*-BuOK formed intermediate amine and released CO gas. This intermediate amine then reacted with **160** to provide intermediate **161**. Finally, reductive elimination of **161** gave *N*-aryl amine product **158** via formation of a new C–N bond and intermediate **159** was recovered.

### 2.6. [3+2] Cycloaddition

1,3-Dipolar cycloaddition, which is defined as the combination of a 1,3-dipole with a multiple bond or bond system called dipolarophile, is a widely applied method for synthesizing heterocycles [100,101,102]. 

Among 1,3-dipolar cycloaddition reactions, [3+2] cycloadditions have been used extensively for the synthesis of pyrrolidine derivatives and other five-membered heterocycles in an efficient way [102,103]. These reactions provide many advantages such as high regioselectivity, high stereoselectivity, and generating multiple stereocenters in one step [101,104]. 

In 2017, Jasiński and co-workers reported the catalyst-free [3+2] cycloaddition of N-methylazomethine ylide with nitroalkenes [105]. Reaction of sarcosine **162** and paraformaldehyde in benzene at 80 °C generated intermediate *N*-methylazomethine ylide **163**. Using [3+2] cycloaddition of in situ formed the intermediate **163** with (2*E*)-3-aryl-2-nitroprop-2-enenitriles **164**, and the desired pyrrolidine was immediately synthesized. Pyrrolidine derivative **165a** was smoothly produced from (2*E*)-3-phenyl-2-nitroprop-2-enenitrile and **162** with 82% yield. Moreover, nitroalkenes bearing methyl and bromo groups on benzene ring were well tolerated with the reaction conditions to afford pyrrolidine products **165b** and **165c** with 76% and 84% yields, respectively (Figure 41). 

In 2020, Chen and co-workers conducted a catalyst-free [3+2] cycloaddition for the synthesis of 3-pyrroline derivatives [106]. Reactions between *o*-hydroxyaryl azomethine ylides and electron-deficient alkynes were carried out in water at reflux without any catalyst. Several pyrrolines **168a**–**168d** were successfully produced with 69–73% yields through the reaction of alkynyl ketones and *o*-hydroxyaryl azomethine ylides. In addition to alkynyl ketones, alkynyl esters were used in [3+2] cycloaddition reactions with *o*-hydroxyaryl azomethine ylides to afford the desired pyrroline derivatives **168e**–**168h** in moderate to high yields (Figure 42). 

[3+2] Cycloaddition was also applied for the synthesis of spirobipyrrolidines from imino esters and 4-benzylidene-2,3-dioxopyrrolidines by Fukuzawa and co-workers in 2022 [107]. [3+2] Cycloaddition reactions were catalyzed by Ag/(*R*, *S*_p_)-ThioClickFerrophos (TCF) in the presence of Et_3_N as a base in THF at 0 °C. A variety of imino esters bearing different benzene and thiophene moieties and many 4-benzylidene-2,3-dioxopyrrolidines were compatible with the reaction conditions, resulting in the formation of desired spirobipyrrolidines in high to quantitative yields with high stereoselectivity for unusual exo’-products (Figure 43). 

Additionally, [3+2] cycloadditions have been utilized for the synthesis of various five-membered heterocycles. For instance, organocatalyzed [3+2] cycloadditions of salicyaldehyde-derived azomethine ylides and nitroalkenes afforded a number of pyrrolidines [108]. In another study, reactions of azomethine ylides with different dipolarophiles catalyzed by (*R*)-DM-SEGPHOS–Ag(I) complex in p-xylene was employed for the preparation of pyrrolidines and pyrrolizidines in high yields and high enantioselectivities [109]. Pyrrolidine azasugar derivatives were prepared via asymmetric [3+2] cycloadditions of azomethine ylides and β-silyl acrylates in the presence of Cu(I) complex Cu(CH_3_CN)_4_BF_4_ [110]. Furthermore, Cu(II)-catalyzed asymmetric 1,3-dpolar cycloaddition of azomethine ylides and α-fuoro-α,β-unsaturated arylketone dipolarophiles yielded chiral 4-fluoropyrrolidines containing four contiguous stereogenic centers [111]. 

### 2.7. Intramolecular Cyclization

#### 2.7.1. Intramolecular C–N Coupling Reaction

In 2013, Sarpong and co-workers reported a one-pot intramolecular C(sp^3^)-N coupling reaction to afford azacycles [112]. The intramolecular reactions were carried out in the presence of *n*-BuLi, ZnCl_2_, and I_2_ in THF (Figure 44). Various *N*-alkyl-2-methylbenzylamine derivatives were employed as substrates for the reaction and they were successfully transformed into the corresponding azacycles. Reactions using substrates bearing a tertiary amine group and phenyl group provided products (**174a** and **174b**) in 53% and 51% yields, respectively. The substrate bearing a bulky adamantyl group, a useful moiety in drug synthesis, was also tested and the target product **174c** was prepared with 47% yield. Using the reaction method, azacycle **174d** bearing a methoxy group at the *ortho* position was successfully obtained with 52% yield under the same reaction conditions, even though methoxy favored lithiation at the *ortho* position of benzenoid. Additionally, syntheses of *N*-acylated isoindoline **174e** and *N*-alkyl isoindoline-1-one **174f** were achieved with high yields. Moreover, six- and seven-membered azacycles (**174g** and **174h**) were successfully prepared with high yields (64% and 74%). 

#### 2.7.2. Intramolecular C–N Amination and Cyclization

In 2020, Du Bois and co-workers developed a two-step process for the synthesis of azacycles involving C–H amination and intermolecular cyclization [113]. In the first step, reactions of alkyl bromide (or alkyl mesylate) with phenyl sulfamate were carried out in the presence of PhI(OPiv)_2_, Al_2_O_3_ as an additive, and [Rh_2_(esp)_2_] as a catalyst in *t*-BuCN for 6 h. Subsequently, the intramolecular cyclization using K_2_CO_3_ in DMF was conducted to form azacycles (Table 3). A wide range of saturated cyclic amines having four-, five-, six-, and seven-membered rings (**177a**–**177d**) was smoothly formed via C–H amination and cyclization reactions. Alkyl bromide substrates containing heterocycle and tertiary carbon were well tolerated in the process and were converted to the corresponding products (**177e** and **177f**) with high yields. Noticeably, this study showed that the *N*-Boc protecting group, which is sensitive to basic conditions, was not decomposed during the process. Substrates bearing dioxolanes were converted to the azacycle product **177h** at a yield of 91%, which could be deprotected for further structural modifications. Importantly, the stereochemistry of the starting materials was preserved during the operation, suggesting that this method would be convenient for the highly efficient synthesis of asymmetrical compounds (**177f**, **177h**). This study also demonstrated that phenoxysulfonyl was a good protecting group for amines and could be deprotected in high yields. 

#### 2.7.3. Intramolecular Cyclization of Dihaloalkanes

In 2014, Gong and co-workers developed a method for the intramolecular cyclization of dihaloalkanes using NiI_2_, Zn, and ligands (2,2′-bipyridine or 4,4′-dimethyl-2,2′-bipyridine) in dimethylacetamide (DMA) to synthesize azacycles (Table 4) [114]. This process successfully cyclized a variety of dibromide and diiodine amines to give azacycles with five-, six-, and seven-membered rings. Cyclization of dihaloalkanes bearing acyl and aryl groups readily afforded pyrrolidine derivatives (**180a**–**180h**) with moderate to high yields. However, *N*-tosyl dibromide was not converted to the corresponding product **180e** due to the electron withdrawal effect. Unsymmetrical dibromide amines and symmetrical secondary dibromide amines were also employed in the reaction, providing branched alkyl pyrrolidines (**180i**–**180k**) with 46–71% yields. Synthesis of larger size azacycles such as six-membered and seven-membered azacycles (**180l** and **180m**) was also achieved, although with lower yields, suggesting a kinetically favorable pathway. 

Previous studies by Gong and co-workers suggested that the formation of organozinc reagents was not involved in the cross-coupling reaction of alkyl halides [115]. A plausible mechanism was proposed, as shown in Figure 45. Ni(0) was combined with substrate **178** to generate X-R_alkyl_-Ni(II) complex **181**. In the presence of Zn, **181** was then reduced to X-R_alkyl_-Ni(I) complex **182**, which further underwent cyclization to form cyclic R_alkyl_-Ni(III)-X complex **183**. Reductive elimination of **183** generated cyclic product **180** and gave Ni(I), which was then reduced to Ni(0) by Zn. Alternatively, the intermediate **181** could also undergo a radical pathway in the presence of the Zn/Ni complex to form radical complex **184** and then this underwent a self-closing ring process to generate **183**.

#### 2.7.4. Intramolecular Cyclization of Diallyl Compounds

In 2013, Chirik and co-workers reported the synthesis of *N*-substituted pyrrolidines via an iron-catalyzed cyclization reaction [116]. In this study, diallyl-tert-butylamine (or diallylaniline) reacted with a bis(imino)pyridine iron dinitrogen complex (^iPr(TB)^PDI)Fe(N_2_)_2_) in benzene-d_6_ under a hydrogen atmosphere (Figure 46). Diallyl amines bearing phenyl and tert-butyl groups were tolerated with the intramolecular [2π + 2π] cycloaddition to afford azabicyclo[3.2.0]heptane derivative **186** in quantitative yields. For the intramolecular hydrogenative cyclization of enynes, the products were dependent on the reaction time. Unsaturated product **188a** was prepared in 30 min with 81% yield, while prolonging the reaction time (3 h) produced saturated pyrrolidine derivative **188b** with 99% yield. Additionally, diyne was employed in the process. However, several byproducts of unsaturated pyrrolidine **190a** were generated during the operation. 

The mechanism of intramolecular cyclization was proposed based on studies on the electronic structures of iron catalyst and metallacycle complexes as shown in Figure 47. Dinitrogen in catalytic complex **191** was replaced by substrate **187** to give intermediate bis(imino)pyridine iron complex **192**. Then, **192** was hydrogenated to provide intermediate **193** by cleaving the C–Fe bonding. Finally, **193** was reduced by nitrogen to generate azacycle product **188** and complex **191** was recovered. 

In 2015, Chirik and co-workers continued to investigate the intramolecular [2π + 2π] cycloaddition of α,ω-dienes for the synthesis of azabicyclo[3.2.0]heptane [117]. Reactions were carried out in the presence of bis(imino)pyridine cobalt dinitrogen derivatives (^iPr^PDI)CoN_2_ or (^Tric^PDI)CoN_2_ in toluene (Figure 48). Reaction of diallyl amines successfully produced several azabicyclo[3.2.0]heptane products bearing trityl, tert-butyl, and 4-fluorophenyl moieties (**195a**–**195c**). Notably, *N*-trityl azabicyclo[3.2.0]heptane **195a** was synthesized from *N*-trityl diallyl amine in excellent yield by treating it with 2.5 mol % (^iPr^PDI)CoN_2_ and 1 mol% of (^Tric^PDI)CoN_2_ in 6.5 h. Similarly, *N*-tert-butyl and *N*-4-flurophenyl azabicyclo[3.2.0]heptane products (**195b**, **195c**) were readily prepared using 1 mol% (^Tric^PDI)CoN_2_ in a short time. However, in the reaction of *N*,*N*-diallylbenzylamine, (^iPr^PDI)FeN_2_ was decomposed. Therefore, using 1 mol % [Fe] catalyst did not give any product. Increasing the amount of [Fe] catalyst to 3% gave the product **195d** with 93% yield. On the other hand, (^iPr^PDI)CoN_2_ and (^Tric^PDI)CoN_2_ remained stable and smoothly gave azabicyclo[3.2.0]heptane product **195d** with 80% and 67% yields, respectively. 

By using in situ EPR spectroscopic monitoring, deuterium labeling and studies on steric and catalyst effects, a proposed mechanism of this reaction is shown in Figure 49. Diallyl amine replaced the dinitrogen of the complex **198** reversibly, forming intermediate **199**. Coordination of the second alkene generated cobalt diene complex **200**. Complex **200** then underwent oxidative cyclization to give complex **201**. Reductive elimination of **201** provided azabicyclo[3.2.0]heptane **195** and recovered the initial catalyst complex **198**. 

#### 2.7.5. Mitsunobu Cyclodehydration Reaction

In 2018, Jones and co-workers developed cyclization of aminoalcohols for the synthesis of *N*-aryl azacycles via the Mitsunobu reaction [118]. This reaction was carried out in the presence of triphenylphosphine and di-tert-butylazodicarboxylate (DTBAD) with or without acetic acid in THF at 0 to 25 °C (Figure 50). The effect of pKa on this reaction was evaluated for the cyclization in the presence of acetic acid as a 5′-OH activator. Various aryl-substituted amino alcohols were tolerated with the reaction, affording *N*-aryl five- and six-membered cyclic amines (**203b**, **203c**, and **203f**) in moderate yields. However, in the absence of acetic acid, cyclic amines **203a**, **203d**, and **203e** were not synthesized due to the high pKa of the amine group (pKa > 15). 

#### 2.7.6. Prins Cyclization

In 2009, Padrón and co-workers reported an iron-catalyzed Prins cyclization process to synthesize azacycles [119]. Homoallyl (or homopropargyl) *N*-tosyl amines were reacted with aldehyde in the presence of FeCl_3_ or Fe(acac)_3_ as a catalyst and trimethylsilyl halides (TMSX) as a halogen source in the corresponding halogenated solvent at room temperature to achieve cyclization (Table 5). In alkyne-Prins cyclization of homopropargylic derivatives, FeCl_3_-catalyzed Prins cyclization of 4-(tosylamino)-1-butyne **204** with several aldehydes **205** bearing isobutyl, cyclohexyl, and benzyl groups in the presence of TMSCl successfully afforded the corresponding chloro-substituted unsaturated azacycles (Table 5, entries **1**–**3**) with 65–80% yields. Additionally, when TMSBr was employed, the bromo-substituted products (Table 5, entries **4**–**6**) were readily formed with 81–88% yields. Replacement of FeCl_3_ by Fe(acac)_3_ did not cause a significant change of the reaction yield for the synthesis of azacycles, where 6-benzyl-4-chloro-1,2,3,6-tetrahydropyridine and 4-bromo-6-butyl-1,2,3,6-tetrahydropyridine (Table 5, entries **7** and **8**) were prepared with 70% and 85% yields, respectively, via reaction using Fe(acac)_3_.

For the Prins cyclization of homoallyl tosyl amines, saturated substituted *N*-tosyl piperidines **208** and **209** were readily produced in the presence of the corresponding iron halide salts (FeCl_3_ or FeBr_3_) with high yields (Table 6, entries **1**–**4**). It is noteworthy that *trans*-pyrrolidine **208** was the major product in all of the experiments. Utilization of Fe(acac)_3_ catalyst increased the reaction efficiency, limiting byproducts while maintaining high yield (Table 6, entry **5**). Moreover, in the reaction using Fe(acac)_3_ catalyst, products bearing alkene, BnO(CH_2_)_2_, and isobutyl (Table 6, entries **6**–**8**) could be prepared at good yields. 

A mechanistic pathway of the reaction was proposed as shown in Figure 51. Aldehyde **205** was activated by iron salt FeX_3_ to form intermediate **210**. Substrate **204** attacked the carbonyl group of **210** to give intermediate **211**. Due to the high stability of iron oxide and nitrogen counterpart in **211**, additionally, FeX_3_ as the only halide source, an indirect way via ligand exchange of FeX_3_ group of **211** with trimethylsilyl halide was needed. Then, **211** interacted with trimethylsilyl halide to provide intermediate **212** and return FeX_3_. Finally, **212** underwent Prins cyclization to form the six-membered azacycle product **206** and HOSiMe_3_ was released. 

The preparation of nitrogen-containing heterocycles through Prins reactions between *N*-sulfonyl homoallylamine and aldehyde or ketone in the presence of AlCl_3_ and trimethylsilyl halide in dichloromethane was reported by Li and co-workers in 2016 (Table 7) [120]. Phenylsulfonamide and its derivatives with electron-donating substituents on a benzene ring were successfully reacted with 4-methylbenzaldehyde to afford *N*-arenesulfonyl azacycles with high yields (78–88%) and with higher diastereoselectivity for *trans*-products. Various substrates with sulfonyl groups were evaluated for the process. Phenylsulfonamide derivatives bearing electron-donating groups on the benzene ring were readily reacted with 4-methylbenzaldehyde to give azacycles with high yields and higher diastereoselectivity for *trans*-products (Table 7, entries **1**–**3**), while reaction using a phenylsulfonamide derivative bearing electron-withdrawing group (Table 7, entry **4**) and methanesulfonylamide (Table 7, entry **5**) yielded the corresponding products with moderate yields. 

A wide range of aldehydes was well tolerated with Prins reactions using halide sources such as TMSCl, TMSBr, TMSI, and BF_3_·Et_2_O (Table 8). Reaction of aryl aldehydes bearing electron-donating groups such as alkyl and methoxy groups, and electron-withdrawing groups such as halogens, trifluoromethyl, nitrile, nitro, and carbonyl groups with TMSX (TMCl, TMSBr, TMSI) gave the desired products with high yields. This study showed that *trans*-products were favored over *cis*-products. However, when BF_3_·Et_2_O was employed for the reaction, lower diastereoselectivity was observed (Table 8, entry **8**).

*N*-tosyl homoallylamine reacted with ketones in the presence of AlCl_3_ and TMSBr to afford the corresponding products with moderate yields (Table 9), while the employment of TMSCl or TMSI did not produce successful results. 

A proposed mechanism for this reaction is presented in Figure 52. In the presence of a Lewis acid, two *E*-, *Z*-conformations of iminium ions could co-exist. However, the *Z*-iminium ion was unstable due to the steric hindrance between the tosyl group and R group. Therefore, the reaction through the formation of the *E*-iminium ion was more favored [121]. The iminium ion was intramolecularly cyclized to form a six-membered ring cation **221** (or **222**) with an equatorial Ts group and an axial R group. Then, nucleophilic attack of the halide ion to cation led to the generation of products. Notably, the steric hindrance of the R group made the *trans*-product the major product. 

## 3. Conclusions

In summary, azacycles, nitrogen-containing heterocycles, play a major role in organic and medicinal chemistry due to their frequent occurrence in various areas including the structures of natural products and FDA-approved drugs. Therefore, the development of efficient synthesis processes to introduce azacycle moieties into small and large molecules has been attractive to chemists. 

As we have shown in this review, numerous synthetic methods for *N*-substituted azacycles have been developed by research groups in recent decades, such as alkylation, *N*-heterocyclization, reductive amination, cross-coupling, and intramolecular cyclization. These methods tolerate a wide range of starting materials with good selectivity, which could be applied for the preparation of useful azacycle compounds. In addition, the starting materials and reagents used in these reactions are commercially available or can be easily prepared. 

Although significant advances have been achieved, some problems remain for scientists to solve. For example, the cost of catalysts, long reaction time, unclear mechanism, low reaction yields, and incompatible substrates are important factors to be considered in future studies. Better understanding of the reactivity, selectivity, and mechanism of these transformations is desperately needed to expand the reaction scope of substrates. In addition, gaining control over the stereoselectivity of these reactions will assist scientists in the synthesis of important bioactive compounds with many chiral quaternary centers. 

The previously reported reactions and catalysts should be further studied for their applications for the synthesis of various azacycle compounds, which are vital to many fields including medicinal chemistry. Moreover, these reagents might be potential reagents for other chemical reactions. Further attempts in the development of novel synthesis of azacycles would provide a powerful toolbox for organic synthesis in the future. We believe that this review will provide an overall picture of recent progress in the synthesis of azacycle compounds.

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
