# Peer review of "Recent Advances in Synthetic Routes to Azacycles"

_molecules, 2023, doi:10.3390/molecules28062737_

Round 1
Reviewer 1 Report
Line 13 change method to methods
Lines 14-15 Change to: In this review, we summarize recent approaches for preparing azacycles from different methods as well as describing plausible reaction mechanisms.
Line 44 Change to: ...expensive additives, organic solvents,...
Lines 63 and 72 Change to: ...anilines with halides...
Line 88 Change to: ...are often hydrolysed... (delete be)
Lines 108-109 delete ... for the hydro-gen-borrowing reaction
Line 125 Change to: ... was applied for the synthesis of monosubstituted...
Line 153 Change to: ... well tolerated with this method...
Line 170 Change to: ...via hydrosilylation reaction. The word via should be in italics here and in all places in the manuscript.
In Scheme 8 the catalyst is Fe(OTf)3 Correct scheme
Line 207 Change to: ... silylated by phenylsilane...
Line 222 give the full name of the acronym MSA
Line 236 Change to: ...which was reduced by H2 to afford...
Line 240 Change to: ...hydrogenation of diester and reaction with aniline.
Line 243 Change to: ...carried out for the synthesis of...
Line 248 yields (71-72%) shown in Scheme 13
Scheme 13 structures 57f-g do not correspond to described products (check structures)
Line 292 68% and 76% yield shown in Scheme 17
Line 304 Change to: ...followed by HCl elimination and formation of...
Line 355 yields (82-90%) are shown in Table 2
The structure of 104g is not showing Me group. Correct it
Line 417 Change to: ... produced the product complex 115 which after elimination of B(O)F2 gave the desired product.
Lines 425, 427 and 452 Change to: hydrogen iodide
Line 433 Change to: 1-(2,6-dimethylphenyl)pyrrolidine 118d).
Line2 472-473 Change to: ...in the presence of POCl3 and DBU (they are not catalysts as they participated in the reaction and are chemically modified)
Line 480 Change to: Plausible mechanism of POCl3 and DBU reaction of arylamine with THF.
Line 496 give the full chemical name of TMP and line 497 give the name of DCE
Line 498 ...give aza (Scheme 32). Which compound??
Lines 503 and 521 ....(thiomorpholine... ) structure of the product does not appear on Schemes 32 and 34. Correct Schemes 32 and 34.
Scheme 35 and its legend does not represent cyclic amines. (Structure 139 represents a primary amine). Correct both
Line 558 give the full name of the acronym HFIP.
Line 566 ...variety of N-aryl piperidines such as esters, amines, alcohols, halogens, azide, sulfonamide, and carbonyl at position C3 or C4 (149e, 149f) were successfully synthesized... Not all structures of such products are presented in Scheme 37. Correct Scheme 37 accordingly to show all possible products mentioned in the text.
Scheme 39 Products 158c, 158f, 158h, 158i, and 158j are not shown in the corresponding structures. Correct Scheme 39 structures accordingly.
Scheme 41 structure 163 what is n ( state range on scheme)
Line 654 give the full name of the acronym DMA
Line 763 BnO should be BnO)CH2)2- . The iodo- group on Table 6 Enrty 8 is shown as i-Bu. Correct table 6.
Line 781 delete ...and...
Lines 788-791 ...,while reaction using phenylsulfonamide derivatives bearing electron-withdrawing groups and alkylsulfonylamide yielded the corresponding products (Table 7, entries 4-5) with moderate yields. Entry 5 on Table 7 shows a methyl group which is not an electron-withdrawing group. Correct Table 7.
On Scheme 49 the cis and trans products depict identical structures. Correct structures to show the right conformations.
Line 811 Change to: ...homoallylamine reacted with...
Line 817 Change to: ... was more favored.
Line 832 Change to: ... could be applied for the preparation of...
Author Response
Thank you for considering our revised manuscript entitled “Recent Advances in Synthetic Routes to Azacycles” for publication in Molecules.
We would like to start by thanking the reviewers for their constructive criticisms and encouraging remarks.
You will find below that we answered all the reviewers’ questions and followed their suggestions by adding more descriptions and explanations.
Reviewer #1’s comments
(Q1) For the comment, “1. "Line 13 change method to methods." ”,
(A1) The word “method” was changed to “methods”.
(Q2) For the comment, “2. "Lines 14-15 Change to: In this review, we summarize recent approaches for preparing azacycles from different methods as well as describing plausible reaction mechanisms." ”,
(A2) The words “to prepare” were changed to “for preparing” and the word “describe” was changed to “describing”.
(Q3) For the comment, “3. "Line 44 Change to: ...expensive additives, organic solvents, ..." ”,
(A3) The words “expensive additive, organic solvent” were changed to “expensive additives, organic solvents”.
(Q4) For the comment, “4. "Lines 63 and 72 Change to: ...anilines with halides......." ”,
(A4) The words “anilines and halides” were changed to “anilines with halides”.
(Q5) For the comment, “5. "Line 88 Change to: ...are often hydrolysed... (delete be).." ”,
(A5) The words “are often be hydrolyzed” were changed to “are often hydrolyzed”.
(Q6) For the comment, “6. "Lines 108-109 delete ... for the hydro-gen-borrowing reaction ..." ”,
(A6) The words “for the hydrogen-borrowing reaction” were deleted.
(Q7) For the comment, “7. "Line 125 Change to: ... was applied for the synthesis of monosubstituted..." ”,
(A7) The words “to synthesize” were changed to “for the synthesis of”.
(Q8) For the comment, 8. "Line 153 Change to: ... well tolerated with this method..." ”,
(A8) The words “well tolerated with the method” were changed to “well tolerated with this method”.
(Q9) For the comment, “9. "Line 170 Change to: ...via hydrosilylation reaction. The word via should be in italics here and in all places in the manuscript...." ”,
(A9) The words “via a hydrosilation reaction” were changed to “via hydrosilation reaction”. The word “via” was changed to italics in all places in the manuscript.
(Q10) For the comment, “10. " In Scheme 8 the catalyst is Fe(OTf)3 Correct scheme..." ”,
(A10) The words “in the presence of Fe(OTf)3 as a catalyst” were changed to “in the presence of Fe(CO)4(IMes) as a catalyst” and the words “Fe(OTf)2 as an additive” were added.
(Q11) For the comment, “11. " Line 207 Change to: ... silylated by phenylsilane. ..." ”,
(A11) The words “silylated by 1-phenylpyrrolidine” were changed to “silylated by phenylsilane”.
(Q12) For the comment, “12. " Line 222 give the full name of the acronym MSA." ”,
(A12) The word “MSA” was changed to “methanesulfonic acid (MSA)”.
(Q13) For the comment, “13. " Line 236 Change to: ...which was reduced by H2 to afford...." ”,
(A13) The words “which reacted with aniline” were changed to “which was reduced by H2”.
(Q14) For the comment, “14. " Line 240 Change to: ...hydrogenation of diester and reaction with aniline...." ”,
(A14) The words “hydrogenation of diester and aniline” were changed to “hydrogenation of diester and reaction with aniline”.
(Q15) For the comment, “15. " Line 243 Change to: ...carried out for the synthesis of...." ”,
(A15) The words “carried out to synthesize” were changed to “carried out for the synthesis of”.
(Q16) For the comment, “16. " Line 248 yields (71-72%) shown in Scheme 13" ”,
(A16) The yields “68-72%” were changed to “70-72%” according to Scheme 13.
(Q17) For the comment, “17. " Scheme 13 structures 57f-g do not correspond to described products (check structures)" ”,
(A17) Description for products 57f-g “2-Methyltetrahydrofuran and 1-naphthylamine and were smoothly employed in reactions of aromatic amines to prepare azacycles (57g, 57f) in moderate yields.” was changed to “2-Methyltetrahydrofuran and 4-fluoroaniline were smoothly employed in this reaction to prepare azacycle 57f in 90% yield. Compound 57g containing napthyl was also synthesized in moderate yield.”.
(Q18) For the comment, “18. " Line 292 68% and 76% yield shown in Scheme 17" ”,
(A18) The yields “72-83%” were changed to “68-76%” according to Scheme 17.
(Q19) For the comment, “19. " Line 304 Change to: ...followed by HCl elimination and formation of......." ”,
(A19) The words “followed by the formation of” were changed to “followed by HCl elimination and formation of”.
(Q20) For the comment, “20. " Line 355 yields (82-90%) are shown in Table 2.." ”,
(A20) The yields “75-92%” were changed to “77-92%” according to Scheme 21.
(Q21) For the comment, “21. " The structure of 104g is not showing Me group. Correct it" ”,
(A21) R2 (methyl group) of the starting material N-methylaniline was eliminated in the reaction of N-methylaniline and tetrahydrofuran to produce compound 104g.
(Q22) For the comment, “22. " Line 417 Change to: ... produced the product complex 115 which after elimination of B(O)F2 gave the desired product..." ”,
(A22) The words “produced the product complex” were changed to “produced the product complex 115 which after elimination of B(O)F2 gave the desired product”.
(Q23) For the comment, “23. " Lines 425, 427 and 452 Change to: hydrogen iodide" ”,
(A23) The words “hydrogen iodine” were changed to “hydrogen iodide”.
(Q24) For the comment, “24. " Line 433 Change to: 1-(2,6-dimethylphenyl)pyrrolidine 118d)..." ”,
(A24) The words “1-(2,6-dimethyl-phenyl)pyrrolidine 118d” were changed to “1-(2,6-dimethylphenyl)pyrrolidine 118d”.
(Q25) For the comment, “25. " Line2 472-473 Change to: ...in the presence of POCl3 and DBU (they are not catalysts as they participated in the reaction and are chemically modified) " ”,
(A25) The words “in the presence of both catalysts POCl3 and DBU” were changed to “in the presence of POCl3 and DBU”.
(Q26) For the comment, “26. " Line 480 Change to: Plausible mechanism of POCl3 and DBU reaction of arylamine with THF..." ”,
(A26) The words “mechanism for POCl3 and DBU-catalyzed reaction” were changed to “mechanism of POCl3 and DBU reaction”.
(Q27) For the comment, “27. " Line 496 give the full chemical name of TMP and line 497 give the name of DCE. ..." ”,
(A27) The words “(TMP = 2,4,6-trimethoxyphenyl)” and 1,2-dichloroethane (DCE) were added.
(Q28) For the comment, “28. " Line 498 ...give aza (Scheme 32). Which compound??.." ”,
(A28) The word “aza” was changed to “the corresponding azacycles”.
(Q29) For the comment, “29. " Lines 503 and 521 ....(thiomorpholine... ) structure of the product does not appear on Schemes 32 and 34. Correct Schemes 32 and 34." ”,
(A29) Structure of compound 137i containing thiomorpholine was added to Scheme 32. The products “137e, 137f, 137g, and 137h” were changed to “137e-137i”. The words “carbonyl groups” were changed to “ester group”.
Structure of compound 141f containing thiomorpholine was added to Scheme 34. The words “piperidines, pyrrolidine, tetrahydroquinoline, morpholine, and thiomorpholine” were changed to “piperidines, pyrrolidine, and tetrahydroquinoline”. The sentence “Utilization of electron-withdrawing groups for the reaction could successfully generate N-aryl cyclic amines.” was removed. The words “N-phenyl piperidine 141d and N-phenyl morpholine 141e” were changed to “N-phenyl piperidine 141d, N-phenyl morpholine 141e, N-phenyl thiomorpholine 141f and 2-methyl-1-phenylindoline 141g.”. Compounds “141g and 141h” were changed to “141h and 141i”.
(Q30) For the comment, “30. " Scheme 35 and its legend does not represent cyclic amines. (Structure 139 represents a primary amine). Correct both”,
(A30) Scheme 35 was changed to represent cyclic amines. The word “aliphatic” in the legend was changed to “cyclic”.
(Q31) For the comment, “31. " Line 558 give the full name of the acronym HFIP." ”,
(A31) The word “HFIP” was changed to “hexafluoroisopropanol (HFIP)”.
(Q32) For the comment, “31. " Line 566 ...variety of N-aryl piperidines such as esters, amines, alcohols, halogens, azide, sulfonamide, and carbonyl at position C3 or C4 (149e, 149f) were successfully synthesized... Not all structures of such products are presented in Scheme 37. Correct Scheme 37 accordingly to show all possible products mentioned in the text." ”,
(A32) Structures of all mentioned compounds were added to Scheme 37. Numbers of compounds were changed according to Scheme 37.
(Q33) For the comment, “33. " Scheme 39 Products 158c, 158f, 158h, 158i, and 158j are not shown in the corresponding structures. Correct Scheme 39 structures accordingly." ”,
(A33) R2 of the starting material 157 was eliminated in the reaction, therefore, products 158a-l did not contain R2. Information about R2 in Scheme 39 was for clarification of starting materials used to synthesize the corresponding products. The word “piperidine”, “morpholine”, and “piperazine” were changed to “piperidines”, “morpholines”, and “piperazines”, respectively.
(Q34) For the comment, “34. " Scheme 41 structure 163 what is n (state range on scheme)..." ”,
(A34) Ranges “X = CH, N” and “n = 0, 1, 2” were added to Scheme 41.
(Q35) For the comment, “35. " Line 654 give the full name of the acronym DMA" ”,
(A35) The word “DMA” was changed to “dimethylacetamide (DMA)”.
(Q36) For the comment, “36. " Line 763 BnO should be BnO)CH2)2- . The iodo- group on Table 6 Enrty 8 is shown as i-Bu. Correct table 6." ”,
(A36) The group “BnO” was changed to “BnO(CH2)2”. The word “iodo-group” was changed to “isobutyl”.
(Q37) For the comment, “37. " Line 781 delete ...and..." ”,
(A37) The words “and in dichloromethane” were changed to “in dichloromethane”.
(Q38) For the comment, “38. " Lines 788-791...,while reaction using phenylsulfonamide derivatives bearing electron-withdrawing groups and alkylsulfonylamide yielded the corresponding products (Table 7, entries 4-5) with moderate yields. Entry 5 on Table 7 shows a methyl group which is not an electron-withdrawing group. Correct Table 7..." ”,
(A38) The words “phenylsulfonamide derivatives bearing electron-withdrawing groups and alkylsulfonylamide yielded the corresponding products (Table 7, entries 4-5) with moderate yields” were changed to “phenylsulfonamide derivative bearing electron-withdrawing group (Table 7, entry 4) and methanesulfonylamide (Table 7, entry 5) yielded the corresponding products with moderate yields.”.
(Q39) For the comment, “39. " On Scheme 49 the cis and trans products depict identical structures. Correct structures to show the right conformations." ”,
(A39) Conformation of the cis product in Scheme 49 (which is now Scheme 52) was corrected.
(Q40) For the comment, “40. " Line 811 Change to: ...homoallylamine reacted with..." ”,
(A40) The words “homoallylamine could react with” were changed to “homoallylamine reacted with”.
(Q41) For the comment, “41. " Line 817 Change to: ... was more favored." ”,
(A41) The words “was favored more” were changed to “was more favored”.
(Q42) For the comment, “41. " Line 832 Change to: ... could be applied for the preparation of..." ”,
(A42) The words “could be applied to the preparation of” were changed to “could be applied for the preparation of”.
We hope that our modifications to the manuscript for the specific concerns and questionable points will satisfy the reviewers and the requirements for the publication of this manuscript.
Reviewer 2 Report
The review article presents recent synthetic methods of saturated azacycles.
The most important suggestion is that authors should replace the paragraph contained in lines 50-51 with the following paragraph "Herein, the present review summarizes recent advances in the synthesis of the following azacycles: azetidine, pyrrolidine, piperidine, azepane etc. The synthetic chemistry publications from the primary literature were retrieved from (add source e.g. the Reaxys database) for the period (e.g. 2000 to 2022)" and provide information regarding the period searched in the database(s) used.
The chemistry presented in this review is sound. The synthetic procedures contain a brief and concise account of the reactants and in many examples a carefully presented plausible mechanism is included. The authors have not presented the synthesis of aromatic azacycles, as expected. If the authors have not used a database such as SciFinder or Reaxys where the structure of the azacycle under investigation was drawn/searched, they will find that a large number of references have been missed out.
On the whole, the manuscript is written in fairly good English. There are however a few grammatical and editorial mistakes that need to be corrected.
The subject matter is of interest primarily to the organic synthetic chemist and to the medicinal chemist. After implementing the provided suggestions to improve the manuscript, publication in Molecules will be possible.
Author Response
Thank you for considering our revised manuscript entitled “Recent Advances in Synthetic Routes to Azacycles” for publication in Molecules.
We would like to start by thanking the reviewers for their constructive criticisms and encouraging remarks.
You will find below that we answered all the reviewers’ questions and followed their suggestions by adding more descriptions and explanations.
Reviewer #2’s comments
(Q1) For the comment, “1. " The most important suggestion is that authors should replace the paragraph contained in lines 50-51 with the following paragraph "Herein, the present review summarizes recent advances in the synthesis of the following azacycles: azetidine, pyrrolidine, piperidine, azepane etc. The synthetic chemistry publications from the primary literature were retrieved from (add source e.g. the Reaxys database) for the period (e.g. 2000 to 2022)" and provide information regarding the period searched in the database(s) used." ”,
(A1) The sentence “Herein, this review summarizes recent advances in various synthetic methods of azacycles.” on page 2 was changed to “Herein, the present review summarizes recent advances in the synthesis of the following azacycles: azetidine, pyrrolidine, piperidine, azepane etc. The synthetic chemistry publications from the primary literature were retrieved from SciFinder for 2013 to 2022”.
(Q2) For the comment, “2. " The chemistry presented in this review is sound. The synthetic procedures contain a brief and concise account of the reactants and in many examples a carefully presented plausible mechanism is included.
The authors have not presented the synthesis of aromatic azacycles, as expected. If the authors have not used a database such as SciFinder or Reaxys where the structure of the azacycle under investigation was drawn/searched, they will find that a large number of references have been missed out..." ”,
(A2) The following sentences for the synthesis of aromatic azacycles were added on page 1. And related references (15 references) were added.
“…In additions, several synthetic methods for aromatic azacycles were reported [46-60].”
Additionally, additional 30 references were inserted into introduction section. And, new writing about [3+2] cycloaddition and 14 references, as well as Schemes 41-43 were added in section 2.6 in page 26-28.
(Q3) For the comment, “3. " On the whole, the manuscript is written in fairly good English. There are however a few grammatical and editorial mistakes that need to be corrected." ”,
(A3) We checked and corrected grammatical and editorial mistakes according to comments.
We hope that our modifications to the manuscript for the specific concerns and questionable points will satisfy the reviewers and the requirements for the publication of this manuscript.
Reviewer 3 Report
Most universal, and high selective protocol for the preparation of pyrrolidines are [3+2] cycloaddition reactions wit hthe participation of azometine ylides and etnene analogs. These-type reaction are realised without presence of any catalyst, under mild conditions and with full atomic economy. Many examples of these-type transfotmation were desribed in the last time [for example: Chemistry of Heterocyclic Compounds, 53, 1161-1162 (2017) Scientiae Radices, 1, 26 (2022)]. Unfortunately, Authors discussed not this strategy on any way. This is evidently weak point of this review which must be improved before the further ecaluation within "Molecules.
Other, minor remarks:
- FDA acronim should be clearly defined.
- Scheme 2:
what is a "microreaction"?
- Scheme 2 and the discussion in the text:
Reaction temperature was specified at 120oC. Its interesting, because the bp ot the solvent is equal 78oC. This issue should be checked and corrected for full clarification.
- Scheme 3:
what is "x" within the formula of th catalyst?
- Scheme 10:
R2 are not specified
- Schemes 11,12,16 (caption)
This is not mechamism but general scheme.
- Scheme 14,18,20,22,24,26,28,30,33,35,40,42,44,46,48,49 and the discussion in the text:
Do the proposed mechanism was supported by any way? This should be commented in the text.
- Scheme 19:
o-Xylene? m-Xylene? p-Xylene? mixture?
- Scheme 38:
Acronim "PC" should be specified. Do the proposed mechanism was supported by any way? This should be commented in the text.
- Scheme 43 and the discussion in the text:
This is not cycloaddition, but electrocyclisation
- Scheme 45
This is not [2p+2p] cycloaddition, but intramolecular [2+2] cycloaddition.
Author Response
Thank you for considering our revised manuscript entitled “Recent Advances in Synthetic Routes to Azacycles” for publication in Molecules.
We would like to start by thanking the reviewers for their constructive criticisms and encouraging remarks.
You will find below that we answered all the reviewers’ questions and followed their suggestions by adding more descriptions and explanations.
Reviewer #3’s comments
(Q1) For the comment, “1. " Most universal, and high selective protocol for the preparation of pyrrolidines are [3+2] cycloaddition reactions wit hthe participation of azometine ylides and etnene analogs. These-type reaction are realised without presence of any catalyst, under mild conditions and with full atomic economy. Many examples of these-type transfotmation were desribed in the last time [for example: Chemistry of Heterocyclic Compounds, 53, 1161-1162 (2017) Scientiae Radices, 1, 26 (2022)]. Unfortunately, Authors discussed not this strategy on any way. This is evidently weak point of this review which must be improved before the further ecaluation within "Molecules." ”,
(A1) Explanation of [3+2] cycloaddition reactions for the preparation of pyrrolidines were added into section 2.6 (between C-N coupling reaction (2.5) and intramolecular cyclization (2.7(new number)). The following sentences and scheme 41-43 were added to explain [3+2] cycloaddition reactions on page 26-28.
“…1,3-Dipolar cycloaddition, which is defined as combination of a 1,3-dipole with a multiple bond or bond system called dipolarophile, has been a widely applied synthetic method for heterocycles [100-102].
Among 1,3-dipolar cycloaddition reactions, [3+2] cycloadditions have been used extensively for the synthesis of pyrrolidine derivatives and other five-membered heterocycles in an efficient way [102,103]. These reactions provide many advantages such as high regioselectivity, high stereoselectivity, and generating multiple stereocenters of in one step [101,104]
In 2017, Jasiński and co-workers reported the catalyst-free [3+2] cycloaddition of N-methylazomethine ylide with nitroalkenes. [105]. Reaction of sarcosine 162 and paraformaldehyde in benzene at 80 oC generated intermediate N-methylazomethine ylide 163. Using [3+2] cycloaddition of in situ formed intermediate 163 with (2E)-3-aryl-2-nitroprop-2-enenitriles 164, the desired pyrrolidines was immediately synthesized. Pyrrolidine derivative 4a was smoothly produced from (2E)-3-phenyl-2-nitroprop-2-enenitrile and 162 with 82% yield. Besides, nitroalkenes bearing methyl and bromo groups on benzene ring were well tolerated with the reaction conditions to afford pyrrolidine products 165b and 165c in 76% and 84% yields, respectively (Scheme 41).
In 2020, Chen and co-workers conducted a catalyst-free [3+2] cycloaddition for the synthesis of 3-pyrroline derivatives [106]. Reactions between o‑hydroxyaryl azomethine ylides and electron-deficient alkynes were carried out in water at reflux without any catalyst. Several pyrrolines 168a-168d were successfully produced in 69-73% yields through the reaction of alkynyl ketones and o‑hydroxyaryl azomethine ylides. In addition to alkynyl ketones, alkynyl esters were also used in [3+2] cycloaddition reactions with o‑hydroxyaryl azomethine ylides to afford the desired pyrroline derivatives 168e-168h in moderate to high yields (Scheme 42).
[3+2] Cycloaddition was also applied for the synthesis of spirobipyrrolidines from imino esters and 4-benzylidene-2,3-dioxopyrrolidines by Fukuzawa and co-workers in 2022 [107]. [3+2] Cycloaddition reactions were catalyzed by Ag/(R, Sp)-ThioClickFerrophos (TCF) in the presence of Et3N as a base in THF at 0 oC. A variety of imino esters bearing different benzene and thiophene moieties and many 4-benzylidene-2,3-dioxopyrrolidines were compatible with the reaction conditions to produce desired spirobipyrrolidines in high to quantitative yields with high stereoseletivity for unusual exo’-products (Scheme 43).
Additionally, [3+2] cycloadditions have been utilized for the synthesis of various five-membered heterocycles. For instance, organocatalyzed [3+2] cycloadditions of salicyaldehyde-derived azomethine ylides and nitroalkenes afforded a number of pyrrolidines. [108] Reactions of azomethine ylides with different dipolarophiles catalyzed by (R)-DM-SEGPHOS−Ag(I) complex in p-xylene was employed for the preparation of pyrrolidines and pyrrolizidines in high yields and high enantioselectivities [109]. Pyrrolidine azasugar derivatives were prepared via asymmetric [3+2] cycloadditions of azomethine ylides and β-silyl acrylates in the presence of Cu(I) complex Cu(CH3CN)4BF4 [110]. Furthermore, Cu(II)-catalyzed asymmetric 1,3-dpolar cycloaddition of azomethine ylides and α-fuoro-α,β-unsaturated arylketone dipolarophiles yielded chiral 4‑fluoropyrrolidines containing four contiguous stereogenic centers [111]..….”
(Q2) For the comment, “2. " - FDA acronim should be clearly defined." ”,
(A2) The words “FDA-approved” was changed to “the United States Food and Drug Administration (FDA)-approved”.
(Q3) For the comment, “3. " - Scheme 2: what is a "microreaction"?" ”,
(A3) In Scheme 2, the word “microreaction” was changed to “flow microreactor”. The symbol “(aq)” was added to clarify the K2CO3 aqueous solution.
(Q4) For the comment, “4. - Scheme 2 and the discussion in the text:
Reaction temperature was specified at 120oC. Its interesting, because the bp ot the solvent is equal 78oC. This issue should be checked and corrected for full clarification." ”,
(A4) In closed vessels of flow microreactor, the reaction mixture could be rapidly heated to temperatures much higher than the boiling point of the solvent under atmospheric pressure in open vessels. [Eur. J. Org. Chem., 2009, 9, 1321-1325; Angew. Chem. Int. Ed., 2004, 43, 6250-6284.]. In this reaction performed by Gao and co-workers, firstly, the precursors were mixed in micromixers, then the reaction mixture was rapidly heated by microwave radiation in sealed vessels. The pressure inside the reactor was controlled by a pressure valve.
(Q5) For the comment, “5. " - Scheme 3: what is "x" within the formula of th catalyst?" ”,
(A5) The catalyst was a heterogeneous and Raney-type catalyst prepared by precipitating the solution containing Cu(NO3)2, Ni(NO3)2, Fe(NO3)3, and Al(NO3)3 with Na2CO3. Structure of the catalyst contained NiO, Cu2O and FeOx, in which NiO and Cu2O were active species for catalysis of the reaction and FeOx was used as the support. The catalyst was denoted as NiCuFeOx by Shi and co-workers. [Chem. Eur. J. 2013, 19, 3665– 3675; ACS Catal. 2020, 10, 311–335]. The letter “x” in NiCuFeOx could indicate different iron oxides generated from the precipitating reaction or the mixture of copper, nickel and iron oxides in general.
(Q6) For the comment, “6. " - Scheme 10: R2 are not specified." ”,
(A6) In Scheme 10, R2 groups were implied in the structures of the products. In particular, compounds 43a-j had R2 as hydrogen. Compounds 43k-l had R2 as the fused benzene ring.
(Q7) For the comment, “7. " - Schemes 11,12,16 (caption), This is not mechamism but general scheme.”,
(A7) The caption of Scheme 11 “Proposed mechanism for the reductive amination of succinic acid and aniline.” was changed to “Proposed reaction pathway for the reductive amination of succinic acid and aniline.”. The sentence “The proposed pathway for this reaction is shown in Scheme 11.”. The sentence “The proposed mechanism of this reaction is shown in Scheme 12.” Was changed to “The proposed pathway of this reaction is shown in Scheme 12.” The caption of Scheme 16 “Possible mechanism for reactions of aniline with DMC and THF in scCO2.” was changed to “Possible reaction pathways for reactions of aniline with DMC and THF in scCO2.”. The sentence “A possible mechanism to give N-aryl cyclic amine is presented in Scheme 16.” was changed to “A possible pathway to give N-aryl cyclic amine is presented in Scheme 16.”.
(Q8) For the comment, “8. " - Scheme 14,18,20,22,24,26,28,30,33,35,40,42,44,46,48,49 and the discussion in the text: Do the proposed mechanism was supported by any way? This should be commented in the text." ”,
(A8) The sentence “Control experiments showed that formation of compound 57 via the transformation of compound 61 in the presence of AlMe3 was achieved to support the mechanism.” was added to 10 page to suggest a mechanism in Scheme 14.
The sentences “Kinetic study of the reaction between 4-fluoroaniline and THF suggested a pseudo-first order reaction with a rate constant of 5 × 10-5 s-1 and an activation energy of 30 kcal mol-1. This activation energy was consistent with the required energy of the proposed mechanism (26.9 kcal mol-1).” were added to 12 page to suggest a mechanism in Scheme 18.
The sentence “Control experiments showed that reaction with TiCl4 alone afforded the desired azacycle with low yield, while the reaction with DBU alone generated no desired product, indicating the essential role of DBU in activating the arylamine to increase reaction yield.” was added to 14 page to suggest a mechanism in Scheme 20.
The sentence “Control reactions of phenylhydrazine in the presence of TiCl4 and TBD at 120 oC and at room temperature showed that the formation of aniline was only detected at 120 oC.” was added to 15 page to suggest a mechanism in Scheme 22.
The words “which was confirmed by isolating and elucidating its structure with crystal X-ray and NMR.” was added to 16 page to suggest a mechanism in Scheme 24.
The sentences “The energy profile of this reaction was similar to TiCl4-mediated reaction and its activation energy (25.7 kcal mol-1) was comparable to that of TiCl4-mediated reaction (26.9 kcal mol-1) [84], therefore, the mechanism was also proposed in a similar way. However, unlike TiCl4-mediated reaction, formation of 114 was the rate-determining step.” were added to 17 page to suggest a mechanism in Scheme 26.
The sentence “This mechanism was supported by three facts including the total inhibition of the reaction by radical inhibitor TEMPO (2,2,6,6-tetramethylpiperidine-1-oxyl), detection of intermediates 121 and 125, and decrease of 125 over reaction time.” was added to 18 page to suggest a mechanism in Scheme 28.
The sentences “This mechanism was confirmed by several facts obtained from control experiments. Phosphoramidic dichloride 129 was only produced when employing both POCl3 and DBU and it was not obtained when employing POCl3 alone or DBU alone. In addition, prepared phosphoramidic dichloride was successfully transformed into the desired product in the reaction with THF, which confirmed the formation of 129 during the reaction.” were added to 20 page to suggest a mechanism in Scheme 30.
The sentence “Formations of intermediates diaryliodonium fluoride and aryl fluoride intermediates were not detected by 19F NMR.” was added to 22 page to suggest a mechanism in Scheme 33.
The sentence “Control experiments showed that this reaction was not affected by adding radical scavenger 1,1-diphenylethylene (DPE), and aryne trap furan, indicating a ligand coupling mechanism.” was added to 23 page to suggest a mechanism in Scheme 35.
The sentences “The mechanism of this reaction was proposed based on several control experiments. Particularly, employing radical inhibitor TEMPO did not affect the reaction yield, indicating a non-radical reaction mechanism. In addition, detection of R2-H by-products confirmed decarbonylation pathway.” were added to 26 page to suggest a mechanism in Scheme 40.
The sentence “Previously reported studies by Gong and co-workers suggested that formation of organozinc reagents was not involved in the cross-coupling reaction of alkyl halides [115].” and a reference were added to 31 page to suggest a mechanism in Scheme 42 (which is now Scheme 45).
The sentence “The mechanism of intramolecular cyclization was proposed as shown in Scheme 44.” was changed to “The mechanism of intramolecular cyclization was proposed as shown in Scheme 44 (which is now Scheme 47) based on studies on the electronic structures of iron catalyst and metallacycle complexes.”.
The sentence “A proposed mechanism of this reaction is shown in Scheme 46.” was changed to “By using in situ EPR spectroscopic monitoring, deuterium labeling and studies on steric and catalyst effects, a proposed mechanism of this reaction is shown in Scheme 46 (which is now Scheme 49).” on 32 page.
The sentence “Due to the high stability of iron oxide and nitrogen counterpart in 211, additionally, FeX3 as the only halide source, an indirect way via ligand exchange of FeX3 group of 211 with trimethylsilyl halide was needed.” was added the text describing Scheme 48 (which is now Scheme 51).
A new reference was added to support the Scheme 49 (which is now Scheme 52) “Therefore, the reaction through the formation of the E-iminium ion was more favored [121].”.
(Q9) For the comment, “9. " - Scheme 19: o-Xylene? m-Xylene? p-Xylene? mixture?" ”,
(A9) The word “Xylene” in Scheme 19 was changed to “o-Xylene”.
(Q10) For the comment, “10. " - Scheme 38: Acronim "PC" should be specified. Do the proposed mechanism was supported by any way? This should be commented in the text.”,
(A10) The words “photocatalyst PC” were changed to “photocatalyst (PC) Ru(bpy)3Cl2” in the paragraph describing the proposed mechanism of Scheme 38. The sentences “Cyclic voltammetry was used to study the redox properties of N-chloropiperidine 150. When adding HClO4, the reduction potential shifted toward positive values which confirmed the SET reduction of N-chloropiperidine 150 upon protonation.” and “UV-vis absorption studies showed that when keeping the mixture of 150, PC and HClO4 in the dark, the mixture absorbed radiation in the blue region while irradiation the mixture with blue light resulted in rapid color change from green orange to green. Thus, the effect of blue radiation was demonstrated.” were added to support the mechanism. The word “arenes” was changed to “arene”.
(Q11) For the comment, “11. " - Scheme 43 and the discussion in the text:
This is not cycloaddition, but electrocyclisation" ”,
(A11) [2π + 2π] cycloaddition to synthesize cyclobutane is thermodynamically favorable but thermally forbidden due to the orbital symmetry constraint [Angew. Chem., Int. Ed. 1969, 8, 781]. However, many attempts have been made to combine π fragments by using photoreactions, activated pi-systems or using transition metals. Chirik and co-workers reported studies on iron-catalyzed intramolecular [2π + 2π] cycloaddition of α,ω-dienes in 2006, and iron-catalyzed intermolecular [2π + 2π] cycloaddition of ethylene and butadiene in 2011 and the mentioned study in 2013 [J. Am. Chem. Soc. 2006, 128, 41, 13340–13341; J. Am. Chem. Soc. 2011, 133, 23, 8858–8861; J. Am. Chem. Soc. 2013, 135, 4862-4877].
Electrocyclization is defined as the thermal or photochemical cyclization of an open conjugated system by forming a σ bond at the ends of conjugated system. Electrocyclization is a reversible conversion. [Tetrahedron Lett. 1965, 6, 1207-1212; Dinda, B. Electrocyclic Reactions. Essentials of Pericyclic and Photochemical Reactions, 2017, PP 13–35]
This reaction by Chirik and co-workers, however, was performed at 23 oC and photochemistry was not involved. In addition, the starting material diallyl amine was also not a conjugated system.
The word “intramolecular” was added to Scheme 46, the captions of Scheme 46 and 47, and in the text for better explaining of this study.
(Q12) For the comment, “12. " - Scheme 45. This is not [2p+2p] cycloaddition, but intramolecular [2+2] cycloaddition." ”,
(A12) The explanation for [2π + 2π] cycloaddition is similar to the answer for question 11 as shown above. The word “intramolecular” was added to the captions of Scheme 48 and 49, and in the text for better explaining of this study.
We hope that our modifications to the manuscript for the specific concerns and questionable points will satisfy the reviewers and the requirements for the publication of this manuscript.
Round 2
Reviewer 2 Report
The authors have satisfactorily revised the manuscript. They have used the Reaxys database and found 60 new references that were added to the Introduction. Significant English grammar and spelling corrections have been implemented.
I believe the manuscript merits publication in Molecules.
Reviewer 3 Report
Authors considered all remarks and substantially improved the manuscript accordingly. In the consequence, i recommend this paper for the publication.